# A Machine Learning-Based Approach for Wildfire Susceptibility Mapping. The Case Study of the Liguria Region in Italy

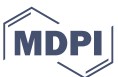

**Marj Tonini** [1,*] , **Mirko D'Andrea** [2] , **Guido Biondi** [2] , **Silvia Degli Esposti** [2] , **Andrea Trucchia** [2] **and Paolo Fiorucci** [2]

[1]  Institute of Earth Surface Dynamics, Faculty of Geosciences and Environment, University of Lausanne, CH-1015 Lausanne, Switzerland

[2]  CIMA Research Foundation, 17100 Savona, Italy; mirko.dandrea@cimafoundation.org (M.D.); guido.biondi@cimafoundation.org (G.B.); silvia.degliesposti@cimafoundation.org (S.D.E.); andrea.trucchia@cimafoundation.org (A.T.); paolo.fiorucci@cimafoundation.org (P.F.)

*  Correspondence: marj.tonini@unil.ch; Tel.: +41-21-692-35-37

**Abstract:** Wildfire susceptibility maps display the spatial probability of an area to burn in the future, based solely on the intrinsic local proprieties of a site. Current studies in this field often rely on statistical models, often improved by expert knowledge for data retrieving and processing. In the last few years, machine learning algorithms have proven to be successful in this domain, thanks to their capability of learning from data through the modeling of hidden relationships. In the present study, authors introduce an approach based on random forests, allowing elaborating a wildfire susceptibility map for the Liguria region in Italy. This region is highly affected by wildfires due to the dense and heterogeneous vegetation, with more than 70% of its surface covered by forests, and due to the favorable climatic conditions. Susceptibility was assessed by considering the dataset of the mapped fire perimeters, spanning a 21-year period (1997–2017) and different geo-environmental predisposing factors (i.e., land cover, vegetation type, road network, altitude, and derivatives). One main objective was to compare different models in order to evaluate the effect of: (i) including or excluding the neighboring vegetation type as additional predisposing factors and (ii) using an increasing number of folds in the spatial-cross validation procedure. Susceptibility maps for the two fire seasons were finally elaborated and validated. Results highlighted the capacity of the proposed approach to identify areas that could be affected by wildfires in the near future, as well as its goodness in assessing the efficiency of fire-fighting activities.

**Keywords:** wildfires; susceptibility mapping; machine learning; random forest; spatial-cross validation

## 1. Introduction

Wildfires represent a hazardous and harmful phenomenon to people and the environment, especially in populated areas. In these areas, the primary cause of ignition is related to human activities. In Europe, for example, 95% of wildfire events are estimated to be induced by humans [1].

Among the weather-induced emergencies, wildfires constitute one of the more complex scenarios. Specifically, in the Mediterranean basin, wildfires are responsible for substantial damaging effects [2]. According to the Advance EFFIS Report on Forest Fires 2017 [3], Italy, Greece, Portugal, and Spain have been affected by a total of 43,733 fires, over a total an area of 89,4244 ha. Despite the financial support of the European Commission and the efforts by local wildfire administrations to prevent and fight wildfires, in the European Union (EU), in 2017, wildfires burnt over 1.2 million ha of natural

lands, causing the death of 127 people among fire fighters and civilians. Over 25% of the total burnt area was in the different Natura 2000 sites, undermining the efforts in preserving natural habitats and key biodiversity for future generations. The losses caused by these fires amounted to 10 billion euros, as estimated by the European Forest Fire Information System [3].

Mapping current and past wildfire events by identifying their location, burned area, starting date, and duration, provides a precious tool for addressing prevention-planning programs aiming to reduce human and material losses. These inventories are an important source of information for the elaboration of hazard, risk, and susceptibility maps. Namely, hazard maps represent the zonation of the spatio-temporal probability of events, while the expected damages or losses are assessed and represented on risk maps. Susceptibility maps do not account for the temporal dimension and indicate zones with a potential to experience a particular hazard in the future, based solely on the intrinsic local properties of a site, expressed in terms of relative spatial likelihood. This definition relies on the basic assumption that future events are expected to occur under similar geo-environmental conditions to those observed in past events. Although these concepts are well-consolidated in the research area related with the risk assessment, especially for landslides [4–7], there is a need for elaborating susceptibility and risk maps for other natural hazards and to develop new quantitative and robust methods supporting their production.

In the field of wildfire risk assessment, fire risk has been defined as a quantitative or qualitative indicator of the likelihood that an area would burn in a certain period of time [8–11]. In this context, modeling fire risks represents a modern tool to support forest protection plans and address fuel management strategies in order to reduce fire consequences [12–14]. More generally, risk and susceptibility analyses are of great importance for land use planning, civil protection, and risk reduction programs. A number of techniques were recently developed to monitor and map the spatial distribution of a burned area and to predict the area at risk of wildfires. These often involve the implementation of physically-based models integrated into a Geographic Information System (GIS), relying on expert knowledge or including statistical analyses and modeling to assess the importance of the predisposing factors [15–24]. Lately, the comparison of deterministic physically-/statistically-based and stochastic approaches highlights the benefit of using data driven methods [25–33] that are able to extract knowledge directly from data. Machine learning has proven to be a mature and reliable tool in quantitative wildfire-related studies, as for the estimation of standing crop and fuel moisture content, often outperforming standard statistical approaches [34–36]. In comparison with deterministic methods, which, given a set of initial conditions (i.e., predisposing factors), always give the same results, stochastic models assume that results obtained through the combination of independent predisposing factors can be slightly different, as a consequence of the randomness of the process.

Although the terms "risk" and "susceptibility" are often used as synonymous, hereinafter we refer to "susceptibility mapping", meaning that the probability for an area to burn is assessed with no consideration of the magnitude of the single wildfire event or of its temporal dimension. Indeed, only the spatial probability for an area to burn in the future, assessed by defining a rank from low to high, is evaluated. This quantitative evaluation is performed considering two aspects: where wildfires occurred in the past, in terms of burned area, and which are the geo-environmental and anthropogenic predisposing factors that favor their spread. In this regard, it is worth noting that meteorological factors, such as wind speed and wind direction, temperature, humidity, and rainfall, are considered as triggering and not predisposing factors. The trigger is a local condition that causes a risk to occur if the area is susceptible to that risk, while the susceptibility is assessed based on predisposing factors, stable over time. The proximity to road and pathway networks and to urban and recreation areas are most frequently mentioned as predisposing human factors for wildfire [37–44]. As regards geo-environmental variables, those related to vegetation type and topography are the most significant drivers, especially in Mediterranean-type regions [45–48].

Italy is particularly affected by wildfires because of the high topographic and vegetation heterogeneity of the territory, as well as the favorable climate conditions that characterize the entire

Mediterranean basin. Wildfires are more frequent and larger in the summer (May–October) than in the winter (November–April) season in almost all Mediterranean countries, since it is hotter and drier. The Liguria region in Italy represents an exception, as it is highly affected by wildfires during the entire year and it sees a number of events and burned areas that can be more important in winter than in summer. Nevertheless, the spatial distribution of burned areas differs in the two seasons, probably due to the vegetation phenology at different altitudes in terms of plant senescence. Therefore, it is important to separately assess wildfire susceptibility in Liguria for the summer and winter season.

In this study, we adopted a stochastic approach based on a machine learning (ML) algorithm to elaborate wildfire susceptibility mapping for Liguria. The ML includes a class of algorithms for the analysis, modelling, and visualization of environmental data and performs particularly well to model environmental hazards, which naturally have a complex and non-linear behavior [49,50]. At present, only a few studies exist in literature on the use of this kind of approach for wildfire susceptibility and risk mapping. Moreover, a rigorous procedure to select the predisposing factors and to validate the results needs to be the established. To fill this gap, we developed different models and we evaluated their prediction performance by considering the effect of: (i) including or excluding the neighboring vegetation type as a predisposing factor and (ii) using an increasing number of folds in the spatial cross-validation procedure.

## 2. Study Area

Liguria has a total area of 5400 km$^2$, found between the Cote d'Azur (France) and Tuscany (Italy) on the northwest coast of the Tyrrhenian Sea. This Mediterranean region is characterized by a complex topography (with a slope that is higher than 40% over 50% of the total area) and dense and heterogeneous vegetation (with more than 70% of its surface covered by forests, see Figure 1).

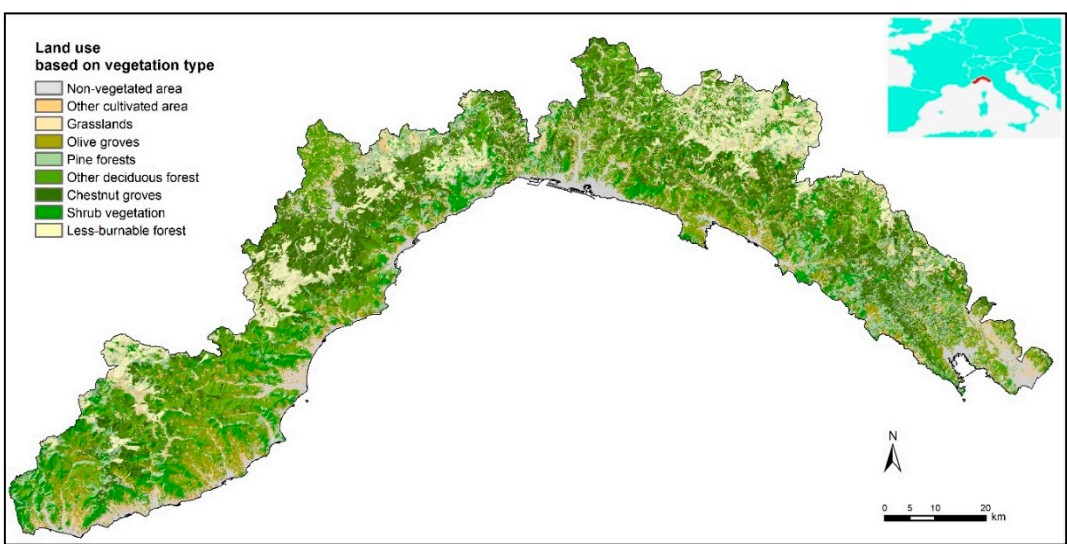

**Figure 1.** Study areas with the location map. Land use results from the aggregation of the original classes represented on the regional map of forest types, provided by the Liguria region.

During the investigated period, 1997–2017, an average of 365 wildfires burnt an area of 55 km$^2$ per year and were a recurrent phenomenon both in the summer and winter. The winter fire regime is mainly due to frequent extremely dry winds from the north in conditions of curing, for most of the herbaceous species, resulting in a number of wildfires and burned areas that can be higher than in summer (Figure 2).

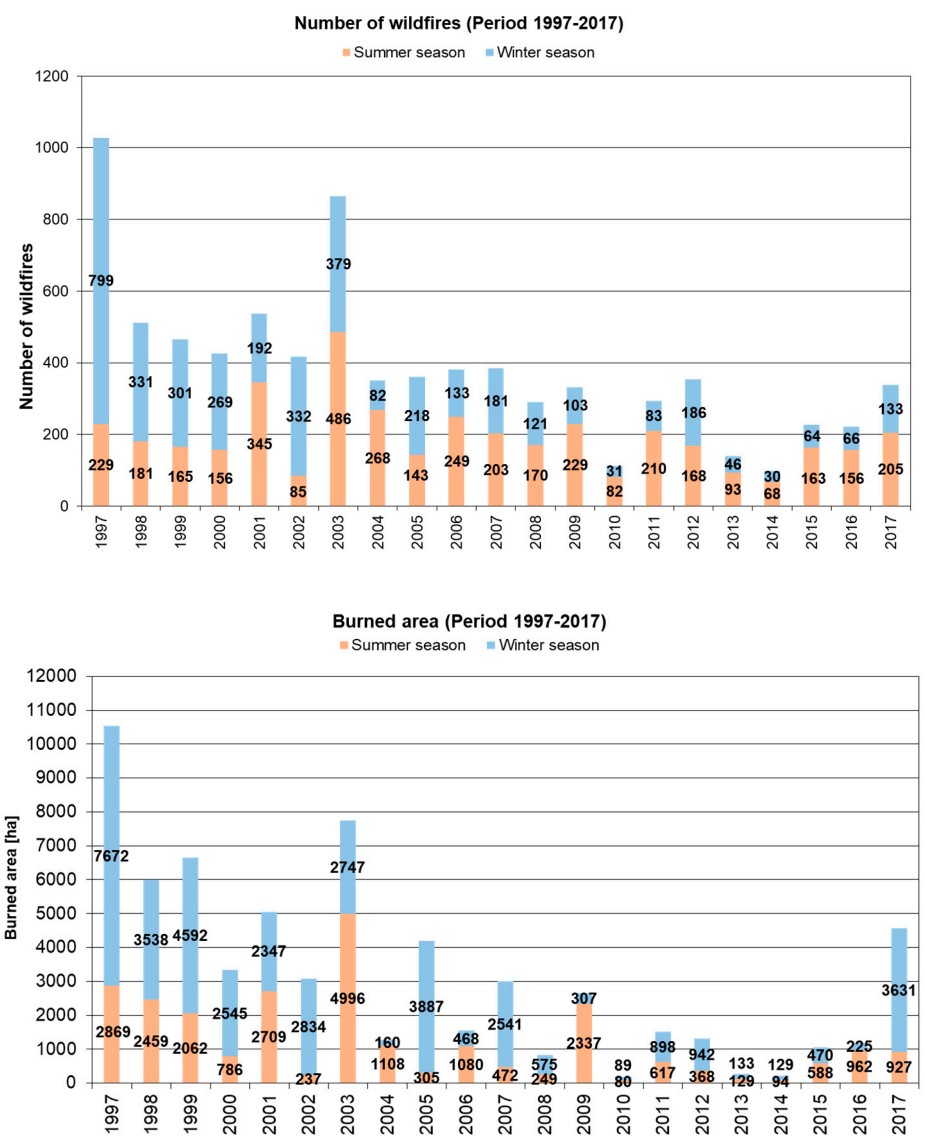

**Figure 2.** Yearly frequency of wildfires (on the top) and burned area (on the bottom) in the Liguria region (Italy) in the last two decades (1997–2017) during the summer and the winter season.

In 1950, the forest cover was limited to 30% of the regional surface, as most of the areas were subject to grazing activities. After World War II, rural communities were engaged in many reforestation programs, using different Mediterranean pine species. Pines and shrub species that are highly flammable spread quickly as agricultural and grazing activities were abandoned. Furthermore, the rural exodus and urbanization that followed the war greatly extended the Wildland–Urban Interface.

## 3. Materials and Methods

### 3.1. The Dataset: Burned Area and Predisposing Factors

The dependent variable of our model, allowing for the assessment of the susceptibility of the region to wildfires, was the burned area available as mapped burned areas and spanning a 21-year period (1997–2017). This dataset was acquired and compiled in shapefile format by the regional forestry service, on the base of GPS-surveying and subsequent digitalization over the cadastral map (scale 1:10,000). It is worth observing that fire perimeters were also affected by the capacity of intervention, which has not been considered in the analysis because of a lack of homogenization of the data.

The seasonality of the fire regime was also considered, partitioning the dataset of burned areas in two macro seasons: winter (November–April) and summer (May–October) (Figure 3)

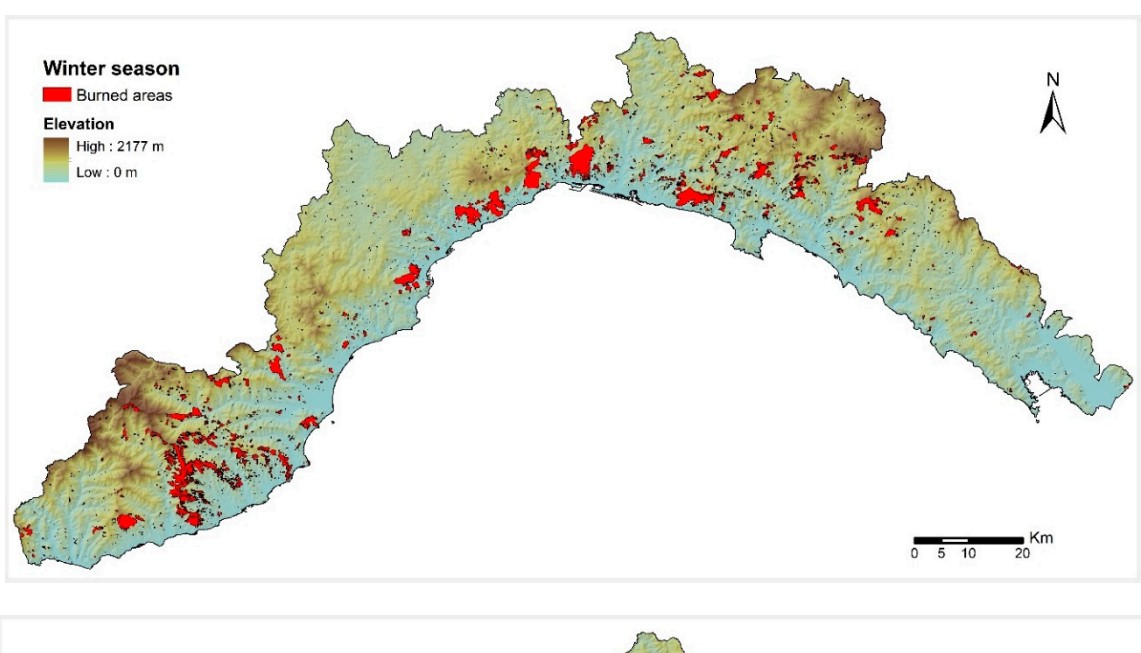

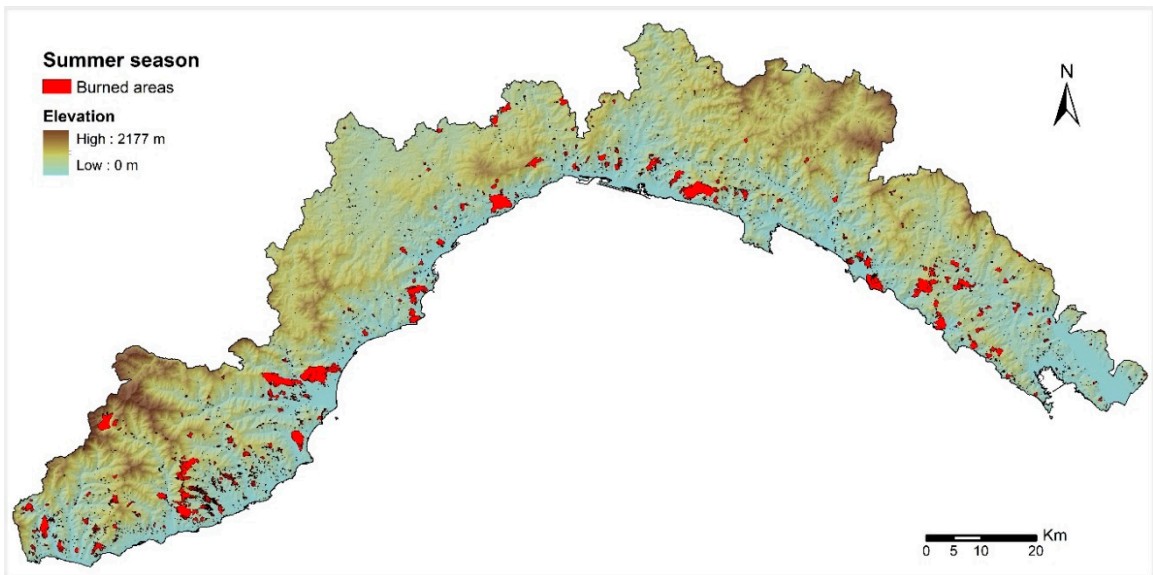

**Figure 3.** Spatial distribution of burned areas during the winter (on the top) and summer season (on the bottom) in the Liguria region (Italy) during the period 1997–2017.

The following independent variables provided detailed knowledge of topography and land cover, allowing for the understanding of the main features involved in wildfire occurrences and their behavior: DEM (digital elevation model) and derivatives (slope, northness, and eastness), distance to anthropogenic features (urban area, road, pathways, crops), protected areas, vegetation type, and neighboring vegetation.

In more detail, northerness and easterness, corresponding respectively to the cosine and to the minus sign of the aspect angle, were considered instead of the pure aspect angle (i.e., the terrain orientation) to avoid the use of a circular variable. The distance values were evaluated by computing, for each pixel, the Euclidean distance to the closest considered element. Protected areas were introduced as a binary variable, computing for each pixel, if it falls inside or outside of these delimited areas. Information on the type of vegetation came from the land use map (scale 1:10,000); furthermore,

the class "forest" was reclassified by splitting the corresponding areal units into the sub-classes identified on the regional map of the forest types (scale 1:25,000). The final map included more than one hundred classes, which, for the purposes of the present study, were aggregated into 37 classes based on the flammability of each vegetation type. The non-vegetated areas, which came from the abovementioned land use map (including urban area, industrial and commercial units, road and rail network, port and airport area, bare rocks, and water bodies), were unified into a unique class under the label "non-flammable area". All these digital layers (Table 1) were provided by the authority of the Liguria region and are available on the official geo-portal (https://geoportal.regione.liguria.it/). The ensemble of the spatial layers, in raster format, were resampled and coregistered (i.e., spatially aligned) to match the same reference image with a spatial resolution of 100 m.

**Table 1.** Independent variables and their characteristics. (*) The neighboring vegetation was only included in the second model for comparison reasons.

| Independent Variables | Acquisition Scale | Variable Type | Range | # of Variables |
|---|---|---|---|---|
| DEM | 1:5000 | Numerical (meters) | 0–2132 | 1 |
| Slope | - | Numerical (degree) | 0–60 | 1 |
| Northness and Eastness | - | Numerical | [−1–+1] | 2 |
| Distance to anthropogenic features | 1:10,000 | Numerical (meters) | 0–9000 | 4 |
| Protected area | 1:25,000 | Binary | 0 or 1 | 1 |
| Vegetation type | 1:25,000 | Categorical | 37 classes | 1 |
| Non-flammable area | 1:25,000 | Categorical | 1 class | 1 |
| Neighboring vegetation * | - | Numerical (percentage) | [0,100] | 38 |

In addition, a set of new variables, taking into account the vegetation cover in the proximity of each pixel, was estimated as the percentage of each classes (including the 37 vegetation types and the non-flammable area) within a 300 by 300 m neighborhood distance. This resulted in 38 additional variables identifying the neighboring vegetation (Table 1). To prove if the neighboring vegetation type enables a rise in the predictive performance of the model, and consequently if this factor needs to be included in this kind of study, two models were tested: the first did not consider the neighboring vegetation (hereafter defined "standard model") and the second did account for this factor (hereafter defined "neighboring vegetation model").

*3.2. Modeling Procedure: Machine Learning Approach*

ML is based on algorithms capable of learning from and making predictions on data, through the modeling of the hidden relationships between a set of input and output variables, representing the predisposing factors (independent variables) and the occurrences of the phenomenon (dependent variable).

A well-established procedure to validate a model is to split the dataset into training, validation, and testing. The "training dataset" is needed to calibrate the parameters of the model, which will be used to get predictions on new data, that is the "validation dataset". Thus, the ultimate purpose of the validation dataset was to provide an unbiased evaluation of the model's fitness. Indeed, a good model is the one which gives accurate predictions on new data and avoids overfitting and underfitting.

Lastly, to provide an unbiased evaluation of the final model and to assess its performance (i.e., generalization), results were predicted over unused observations, defined as the "testing dataset". Model generalization refers to the ability of a model to perform good predictions on new/previously unseen data, drawn from the same distribution as the data used to create the model. This concept is closely related with overfitting: this happens when a model performs well on the training data but is unable to make good predictions on new data. Thus, a model that perfectly fits the training data normally displays a large generalization error.

A binary classification problem, such as the prediction of burned and unburned areas in the present case study, can be solved by counting how many times each observation is classified as "one"

(burning) or "zero" (non-burning) and normalizing the result over the total number of predictions. This provided "probabilistic outputs", the values of which ranged between 0 and 1, which can be used to elaborate susceptibility maps. In these, the value of each pixel represents the probabilistic predicted value of burning in the future. The proposed approach involves the generation of pseudo-absences to ascertain an unburned area. Indeed, to assure a good generalization of the model, avoiding the overestimation of the low classes (i.e., with a probabilistic value close to 0), pseudo-absences need to be generated in all cases where they are not explicitly expressed (e.g., wildfire location is known, normally as mapped burned areas, but the not-burned areas have to be defined). We solved this problem by randomly generating a number of absences equal to the number of presences (i.e., equal to the number of pixels burning).

Analyses were performed using random forest [51], an ensemble ML algorithm based on decision trees. The most important hyperparameters that need to be specified are the number of decision trees (*ntree*) and the number of variables randomly sampled as candidates at each split (*mtry*). As a general rule, *ntree* should be large enough to ensure that every observation (i.e., input rows) gets predicted at least a few times. The standard value for *mtry* in classification problems is equal to the square root of the number of predictors (i.e., the independent variables), but this value can be optimized. Operationally, the algorithm generated *ntree*-subsets of the training dataset by bootstrapping (i.e., random sampling with replacement), each subset counting about two-thirds of the observations. A decision tree was then generated for each subset, considering a reduced number of variables (*mtry*), randomly selected. At each node, the Gini impurity was computed and the variable that minimized this value was selected as the best one for the split. Specifically, Gini impurity represented the probability of classifying incorrectly an observation in the dataset if it were classified randomly, according to the class distribution in the dataset. This process was iterated up to the maximum level and it could stop when each node contained less than a fixed number of data points. For a classification problem, the prediction of new data was finally computed, taking the maximum voting, which can be converted into a probabilistic output. The hyperparameters of the model were optimized by evaluating the prediction-error on those observations that were not used in the training subsets (called "out-of-bag"). In this study, hyperparameters were set to 750 for *ntree* and to the rounded up square root of the number of predictor factors for *mtry*, both optimized by applying a trial and error process.

### 3.3. Model Validation

As mentioned above, a well-established procedure in ML to validate a model is to split the dataset into training, validation, and testing.

When dealing with a spatial environmental phenomenon, if the validation dataset is selected randomly, observations can be located close to the training dataset, leading to an over-estimation of the predictive performance of the model. This circumstance is known as "spatial autocorrelation", meaning that observations close to each other hold similar characteristics. To overcome this issue, training and validation data have to be selected far enough apart in the geographic space. To this end, in the present study, we adopted the spatial k-fold cross validation using an increasing number of folds. Methodologically, the k-fold cross validation consisted of splitting the dataset into k folds, holding out a fold at a time, training the model on the remaining k-1 folds and finally validating the model using the hold out fold. The process was repeated k-times and the evaluation scores resulting from each folding were finally averaged. In the same way, the final prediction value was computed as the arithmetical mean among all the predictions estimated from each folding. We compared three models, considering 1-, 5-, and 9-folds. The space was organized into spatial blocks of $15 \times 15$ km, resulting in 50 blocks covering the entire study area. In the case of 1-fold, the validation dataset was defined by randomly selecting 10 blocks only once, which corresponded to the random selection of the 20% block of observations. In the case of 5- and 9-folds (Figure 4), each single fold included, respectively, 10 blocks (corresponding to the 20% of observations) and 5 or 6 blocks (corresponding to about 10% of the observations).

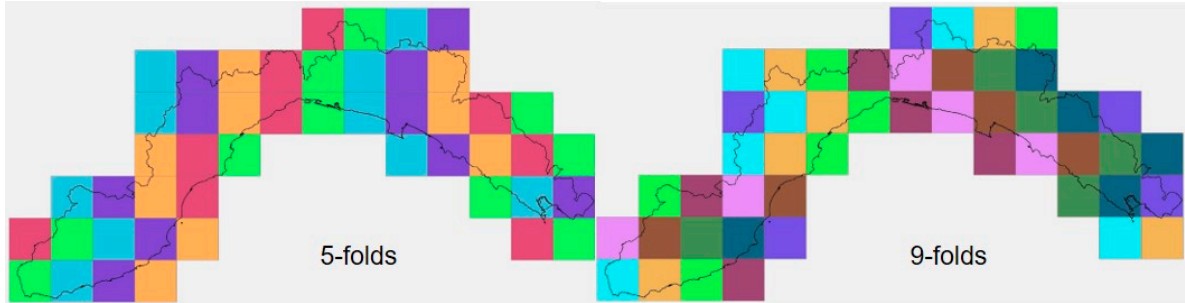

**Figure 4.** Spatial arrangement of the blocks for the 5- and 9-folds adopted in the work (each color corresponds to a single fold).

The testing dataset was defined by splitting the original dataset by years: Mapped burned areas observed in the period 1997–2011 were used in the training procedure, while the last 6 years (2012–2017), corresponding to about 25% of the entire dataset, were held out for testing the predictive performance of the model. This was assessed by computing the percentage of an area with a probabilistic predicted value of burning within a certain range, falling inside the burned areas in the testing dataset (*Testing BA*, Table 2). Finally, the model that predicted better was the one that resulted in a higher percentage for high range values and a lower percentage for low range values. The probabilistic predicted values were ranked into five classes, defined by the following percentile ranking ranges: 25th, 50th, 75th, 90th, and 95th. It is worth pointing out that the 25th percentile represented 25% of the area with the lowest probabilistic predicted values of burning, while the 95th percentile represented the 5% of the area with the highest probabilistic values, and so on for the intermediate ranges. The rank-limits corresponded to different values of the probabilistic output (*Prob_value*, Table 2) resulting from each model, allowing a flexible interpretation of the results. In addition, keeping in mind the goal of delivering unbiased results by adopting different fitness estimators, the root mean square error (RMSE) was also computed (Table 3). The RMSE is based on the difference between the predicted outputs and the observed true values, where observations can only assume the value "one" (if the area burned) or "zero" (if the area did not burn), while the predicted output is a probabilistic value expressed as a floating number between zero and one.

**Table 2.** Validation of the models (see Section 3.2 for more details).

| Winter Season | | 1-Fold Cross Validation | | | | 5-Fold Cross Validation | | | | 9-Fold Cross Validation | |
| --- | --- | --- | --- | --- | --- | --- | --- | --- | --- | --- | --- |
| | | Standard Model | | Neighboring Vegetation | | Standard Model | | Neighboring Vegetation | | Neighboring Vegetation | |
| Classes | Total Area (%) | Testing BA (%) | Prob_Value | Testing BA (%) | Prob_Value | Testing BA (%) | Prob_Value | Testing BA (%) | Prob_Value | Testing BA (%) | Prob_Value |
| 25% | 25 | 5.07 | 0.13 | 3.85 | 0.09 | 4.42 | 0.11 | 3.5 | 0.07 | 3.36 | 0.07 |
| 50% | 25 | 4.95 | 0.25 | 3.4 | 0.22 | 3.44 | 0.23 | 3.27 | 0.18 | 3.17 | 0.17 |
| 75% | 25 | 10.52 | 0.48 | 8.8 | 0.47 | 8.9 | 0.43 | 6.22 | 0.39 | 6.44 | 0.39 |
| 90% | 15 | 17.64 | 0.78 | 14.98 | 0.74 | 15.77 | 0.7 | 13.05 | 0.68 | 11.91 | 0.69 |
| 95% | 5 | 14.38 | 0.91 | 15.26 | 0.87 | 15.67 | 0.85 | 17.7 | 0.83 | 16.43 | 0.85 |
| 100% | 5 | 47.26 | 1 | 52.86 | 1 | 51.78 | 1 | 56.26 | 1 | 58.69 | 1 |
| >75% | | 79.28 | | 83.1 | | 83.22 | | 87.01 | | 87.03 | |
| Summer Season | | 1-Fold Cross Validation | | | | 5-Fold Cross Validation | | | | 9-Fold Cross Validation | |
| | | Standard Model | | Neighboring Vegetation | | Standard Model | | Neighboring Vegetation | | Neighboring Vegetation | |
| Classes | Total Area (%) | Testing BA (%) | Prob_Value | Testing BA (%) | Prob_Value | Testing BA (%) | Prob_Value | Testing BA (%) | Prob_Value | Testing BA (%) | Prob_Value |
| 25% | 25 | 4.71 | 0.08 | 1.04 | 0.04 | 4.04 | 0.06 | 0.8 | 0.04 | 0.8 | 0.04 |
| 50% | 25 | 7.52 | 0.23 | 4.64 | 0.17 | 9.39 | 0.19 | 5.08 | 0.14 | 5.54 | 0.14 |
| 75% | 25 | 17.94 | 0.51 | 18.27 | 0.41 | 15.16 | 0.44 | 19.77 | 0.35 | 18.44 | 0.35 |
| 90% | 15 | 24.45 | 0.78 | 26.19 | 0.7 | 23.31 | 0.69 | 21.51 | 0.65 | 22.11 | 0.66 |
| 95% | 5 | 14.06 | 0.91 | 14.6 | 0.87 | 14.6 | 0.83 | 15.06 | 0.83 | 14.73 | 0.85 |
| 100% | 5 | 30.66 | 1 | 33.43 | 1 | 33.5 | 1 | 37.71 | 1 | 38.31 | 1 |
| >75% | | 69.17 | | 74.22 | | 71.41 | | 74.28 | | 75.15 | |

**Table 3.** Root mean square error based on the difference between the predicted and observed value.

| Winter Season | 1-Fold | 5-Folds | 9-Folds |
|---|---|---|---|
| Standard model | 0.407 | 0.380 | - |
| Neighboring vegetation | 0.377 | 0.354 | 0.351 |
| Summer season | 1-fold | 5-folds | 9-folds |
| Standard model | 0.437 | 0.428 | - |
| Neighboring vegetation | 0.411 | 0.411 | 0.411 |

## 4. Results

The main results of the present study are the following: (i) the models comparison, including the "standard" vs. the "neighboring vegetation" model and the use of 1- vs. 5- vs. 9-folds for the spatial cross validation; (ii) the probabilistic predicted values as the main output of random forest, allowing to elaborate susceptibility maps of wildfires for the winter and summer seasons.

### 4.1. Models Comparison

The values indicated in Table 2 allow comparing and evaluating the predictive performance of different models: the "standard" vs. the "neighboring vegetation" model and the use of 1- vs. 5- vs. 9-folds spatial cross validation for the winter (values above) and summer seasons (values below). In addition to the percentile ranking ranges (i.e., "Classes"), the corresponding probabilistic predicted value (i.e., Prob_value) is specified. The field "Total area" indicates the surface predicted by the model to fall within each range, normalized over the entire area. The field "Predicted BA" represents the fraction of the *"Total area"* falling inside the burned area detected in the testing dataset (i.e., 2012–2017 mapped burned areas). For example, looking at one-fold standard model, the first class (25th percentile) indicates that only 5.07% of the 25% of the area with the lowest probabilistic predicted values of burning (corresponding to <0.13), fell inside the burned area in the testing dataset. On the opposite extreme, the 95th percentile indicates that about 47% of the 5% of the area with the highest probabilistic predicted values (corresponding to >0.91), fell inside the burned area in the testing dataset. Thus, a way to compare the different models is to evaluate which model predicts the largest area with the highest probability of burning falling inside the burned area in the testing dataset (hereafter defined for brevity as "predicted burned area"), and vice-versa for the lower classes. To facilitate this comparison, the last 75th percentile (i.e., 25% of the area with the highest probabilistic predicted value of burning) was considered and discussed in the following. Results show that the neighboring vegetation model performed better than the standard model in both seasons. Indeed, the predicted burned area was higher for both 1- and 5-folds cross-validation when the model included the information concerning the type of vegetation and non-flammable area in the neighboring of each pixel. The increment in this case is of about three percentage points or more. Instead, when comparing 5- with 5-folds cross-validation, the predicted burned area increased y only 0.02% in the winter season and 0.87% in the summer season. So, using 9- instead of 5-folds allows to slightly increase the predictive performance of the model, facing the fact that the training algorithm is much more computationally intensive. On the other hand, the predictive performance of using 5-folds cross-validation compared with the 1-fold increased by about four-percentage points in the winter season, when considering both the standard and the neighboring vegetation model, and was 2.24% and only 0.07% for the equivalent models in the summer season. Despite this last low value, which will be discussed later, overall, using 5-folds cross-validation results in a better predictive performance of the model than selecting a random 20% of the dataset by blocks for validation (corresponding to 1-fold).

The root mean square errors computed for all the models (Table 3) confirm these results. As expected, all RMSEs were lower than 0.5, as, on average, each model performed good prediction of burned and unburned areas. The neighboring vegetation model performed better than the standard model in both seasons. Increasing the number of folds in the cross validation allowed to increase the

model's performance in all the cases, except in the case of the neighboring vegetation model in the summer season, which performed the same, regardless of the number of folds.

All the models performed better in winter than in summer. In the winter, the values of the testing burned area, considering the last 75th percentile, ranged from about 80% (1-fold "standard model") to 87% (5- and 9-folds "neighboring vegetation model"). In the summer, these values were significantly lower, ranging from about 70% (1-fold "standard model") to 75% (5- and 9-folds "neighboring vegetation model"). These performances were confirmed by the RMSE, showing lower values in summer than in winter for the same model.

The neighboring vegetation model using 5-folds cross-validation performed better. This last model was finally evaluated by computing the fraction of predicted burning area above the 75th percentile falling inside the testing burned areas for each single year and by season. Values are expressed both as percentage and number of pixels (Table 4). This allowed for the investigation of the influence of the testing dataset on the predictive performances of the model. In the winter season, the model performed well every year, with values ranging from 83.4% in 2013 to 91.7% in 2016. In the summer season, performances were still quite good in the first five testing periods (years 2012–2016), while, in the last year (2017), the assessed value dropped to 45.7%. At the same time, we noticed that the size of the burned area, which highly differed in each testing period, did not compromise the predictive performance of the model. More specifically, this result seems to imply that something during the 2017 wildfire summer season in Liguria did not work as predicted by the model.

**Table 4.** Model validation evaluated by computing the percentage of the predicted burning area above the 75th percentile falling inside the testing burned areas for each single year (BA >75%). The number of pixels above and below this value for each season are also shown.

| Year | Winter Season | | | Summer Season | | | Tot_Year |
| | BA > 75 % (%) | BA > 75% (# *pixels*) | Tot_Winter (# *pixels*) | BA > 75 % (%) | BA > 75% (# *pixels*) | Tot_Summer (# *pixels*) | (# *pixels*) |
|---|---|---|---|---|---|---|---|
| **2012** | 84.1 | 844 | 1003 | 77.8 | 337 | 433 | 1436 |
| **2013** | 83.4 | 121 | 145 | 86.9 | 140 | 161 | 306 |
| **2014** | 86.0 | 117 | 136 | 92.8 | 103 | 111 | 247 |
| **2015** | 91.2 | 465 | 510 | 84.9 | 535 | 630 | 1140 |
| **2016** | 91.7 | 220 | 240 | 92.7 | 936 | 1010 | 1250 |
| **2017** | 86.4 | 3144 | 3640 | **45.7** | 449 | 983 | 4623 |
| **Tot** | | | 5674 | | | 3328 | 9002 |

## 4.2. Susceptibility Mapping

Random forest gives as an output a probabilistic value, expressing the probability for each pixel to burn in the future under the assumption of a set of predisposing variables. These values were used to elaborate wildfire susceptibility maps in the winter and summer seasons (Figure 5). To this end, we retained the results that were obtained by the model that performed better, which was the neighboring vegetation model using 5-folds cross-validation. The two seasons displayed different behavior in relation to the predicted susceptibility to wildfires.

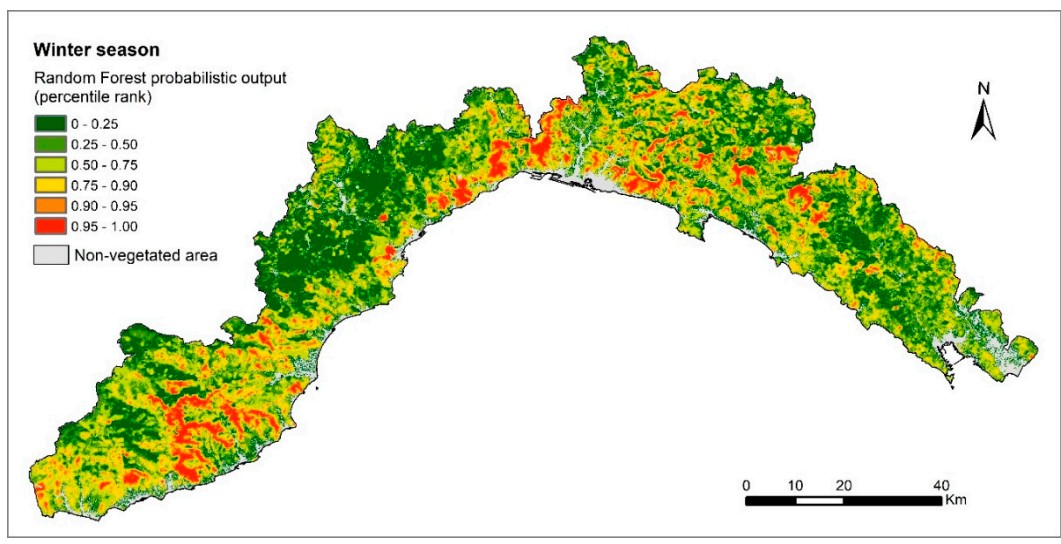

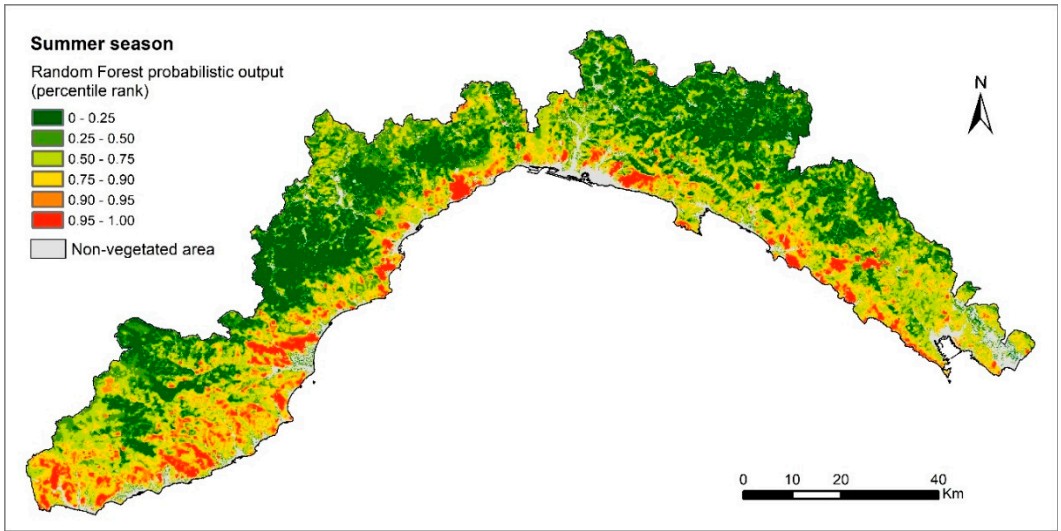

**Figure 5.** Wildfire susceptibility map in the winter (on the top) and summer seasons (on the bottom) for the Liguria region (Italy).

Higher classes (above the 90th percentile) characterized the coastal area in summer and the more elevated inland areas in winter. This could be due to the state of the vegetation: In winter, vegetation is more stressed and senescent at a higher altitude due to the lower temperature in altitude. On the contrary, in summer, vegetation is dryer and more burnable at lower elevations because of the high temperature and the dry weather. Finally, even though the two models implemented separately for the winter and the summer seasons used the same independent variables as the input, random forest succeeded in discriminating among the two patterns in the distribution of the wildfire-susceptible areas, thanks to the training procedure.

## 5. Discussion

In a very recent review paper, Jain et al. [52] found a total of 298 publications, from 1996 to 2019, relevant to the topic of machine learning applications in wildfire science and management, with a very strong increase in the last five years of investigation. Among them, a considerable number of references (71) used various ML algorithms to map wildfire susceptibility. Nevertheless, only a few papers considered time series longer than a few years, allowing to assess the predictive performance of the model in the near future [26,31,53] or separately in summer and winter seasons [54].

In the present paper, we introduced an innovative ML approach, based on random forest, which allowed the elaboration of the wildfire susceptibility map for the Liguria region in Italy. Susceptibility was assessed by evaluating the probability for an area to burn in the future, taking into account the spatial extension of past-burned areas and the geo-environmental factors that favor their occurrence (i.e., DEM and its derivatives, distance to an anthropogenic feature, protected area, vegetation type). An alternative model, including the neighboring vegetation type at each location, was developed and compared with the standard model. For validation purposes, we adopted the spatial k-fold cross-validation (with k = 1, 5, and 9) and then we evaluated the predictive performances of the different models over an independent testing dataset. Finally, we compared the standard vs. the neighboring vegetation model and the use of 1- vs. 5- vs. 9-folds spatial cross-validation, both for the winter and summer seasons. The results show that: (1) the neighboring vegetation model performed better than the standard model in both seasons; (2) 5 represents the optimal number of folds; (3) all the models performed better in the winter season than in the summer season.

The predictive performance of the models implemented in the present study was globally good, as proved by their high capacity of discriminating most of the burned area within the 75th percentile for most of the testing periods (2012 to 2017) and in both the wildfire seasons. This attests the good generalization capabilities of our models. The neighboring vegetation model using 5-folds cross-validation gave the best results in terms of predictive performance and was retained to elaborate the winter and the summer wildfire susceptibility maps. The performance of this model in the summer 2017 testing period manifested the only notable exception to the overall good prediction capability and it will be the object of discussion in the following. Instead of considering this fact a downside of the proposed modeling framework, it shifted the attention towards a specific situation: Fire management in summer 2017. As a matter of fact, it is worth considering that the Italian Forestry Corp, in charge of fire management since 1984, was dismissed at the end of 2016 (DLGS 19/08/2016 n. 177). To counter this, the Liguria region named Italian Fire Fighters (CNVVF) in charge of forest fire management. Before 2017, the role of CNVVF was limited to the management of fire in the Wildland–Urban Interface and was mainly restricted to the safeguard of civilians (protection of houses and infrastructures). In January 2017, a week of severe fire danger caused the burning of about 3000 ha, most of them characterized by high or extreme fire risk. However, the summer season of 2017 was atypical, not only in terms of fire management procedures, but also with respect to the meteorological conditions. This season was in fact characterized by a long drought, but with a remarkable average relative humidity higher than 67%. Only a couple of days were characterized by relative humidity lower than 40%. These meteorological conditions favored the mop up phase, which caused each single fire to be reignited several times, extending the fire propagation for many days.

In the period ranging from 1st May to 31st October 2017, 205 wildfires burned a total area of 927 ha. Only six events burned an area greater than 50 ha each, resulting in 554 ha (corresponding to 60% of the total burned area). These six events were highlighted for a better understanding of the implication that the new management can have on the predictive performance of the model. As is evident in Figure 6, most of the burned area within these six largest wildfires was predicted by the model as belonging to a middle-to-low level of susceptibility. This result reflects that random forest could not take into account the management issues of 2017: The underestimation of the mop up phase. However, it successfully predicted the areas to be burned in the future under unchanged geo-environmental and management conditions, as proven by the good performances of the model in the previous testing periods.

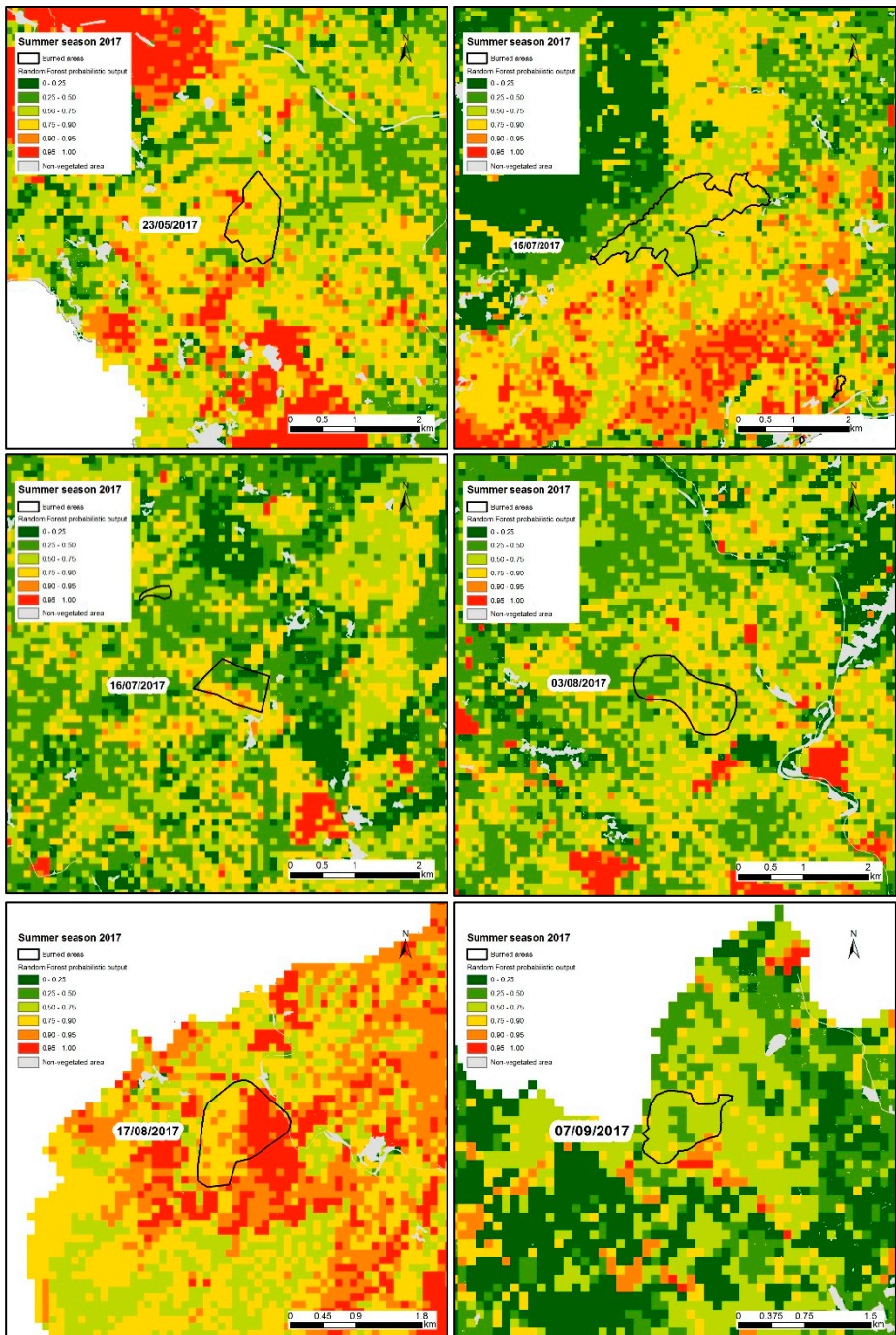

**Figure 6.** The six largest wildfire occurrences of summer 2017, along with the fire susceptibility map.

## 6. Conclusions

The proposed approach proved to be globally effective to deal with large, high dimensional spatial data, which reflects the great flexibility of machine learning in general. In particular, one advantage of random forest is its capability of handling directly categorical variables, such as land use classes or vegetation types.

The importance of considering the vegetation cover in the proximity of each pixel, specifically the type of vegetation, as an additional variable is proved by the results of the present research, and we recommend including this factor in similar studies. Moreover, we recommend the use of k-fold

cross-validation to improve the predictive performance of machine learning-based models seeking to assess the susceptibility of hazardous events in general and wildfires in particular.

Ultimately, the results of the present study highlight the importance of taking into account possible changes of surrounding conditions, which can affect the predictive performance of the model in space and in time. Indeed, despite the rapid capacity of intervention of the Italian Fire Fighters, the fire brigade did not consider the mop up phase, which, in summer 2017, caused each single fire to be reignited several times. These considerations put in evidence the capacity of the proposed approach to identify the efficiency of fire-fighting activities, specifically if fire extinction procedures are handled with different modalities compared with the procedures used previously and implicitly valued in the model. Before 2017, the different tactics of fire management could have resulted in a lower extension of the burned areas, as identified by the models and results from the susceptibility map.

**Author Contributions:** Data curation, G.B. and S.D.E.; Formal analysis, M.T. and M.D.; Funding acquisition, M.T. and P.F.; Investigation, M.T. and M.D.; Methodology, M.T.; Project administration, P.F.; Software, M.D.; Supervision, P.F. and M.T.; Validation, M.T. and M.D.; Visualization, G.B.; Writing—original draft, M.T.; Writing—review & editing, M.T., A.T. and P.F. All authors have read and agreed to the published version of the manuscript.

**Funding:** This research was funded Swiss National Science Foundation (FNS): IZSEZ0_186483.

**Acknowledgments:** The authors acknowledge the Italian Civil Protection Department - Presidency of the Council of Ministers, who founded this research through the convention with the CIMA Research Foundation, for the development of knowledge, methodologies, technologies, and training, useful for the implementation of national wildfire systems of monitoring, prevention, and surveillance. A visiting period of Mirko D'Andrea, from CIMA, at the Institute of Earth Surface Dynamics (UNIL), has been possible thanks to a scientific exchange grant funded by the Swiss National Science Foundation (IZSEZ0_186483). The authors are grateful to Regione Liguria and the Liguria's regional direction of the Italian National Fire Corps for data providing and for the stimulating discussion about the summer fire season 2017. Thanks to the anonymous reviewers for helping to improve the manuscript.

**Conflicts of Interest:** The authors declare no conflict of interest.

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
