# Peer review of "A Machine Learning-Based Approach for Wildfire Susceptibility Mapping. The Case Study of the Liguria Region in Italy"

_geosciences, doi:10.3390/geosciences10030105_

Round 1

Reviewer 1 Report

    General comments:

I have read the manuscript entitled “A machine learning-based approach for wildfire susceptibility mapping. The case study of the Liguria region in Italy”. The article deals with the importance of machine learning tools to map wildfire susceptibility in Italy. The article is interesting as it provides the use of the “Machine learning: Random Forest” method to identify the different variables for mapping wildfire. However, the paper needs more explanation on the gap identification and overall objectives of the study. The abstract needs to rewritten.

The introduction starts with the usefulness of mapping hazardous events but since this paper is more focused on wildfire mapping, the author needs to start with an introductory sentence focusing on the cost and area associated with wildfire?  The author should provide enough literature review on different statistical and stochastic approaches and its implementation on mapping wildfire susceptibility. For example, the author can explain more at the end of the second paragraph. Also, explain why they use machine learning techniques. The author can add as “One of the studies done in the tallgrass prairie in the US, the study revealed that the artificial neural network, a machine learning technique was found better than statistical method as stepwise linear regression method in estimating fuel moisture content (wildfire estimation) using day of year, canopy height and spectral reflectance data [32]”. I have added the bibliography of the paper in the pdf attached on page 2.

About the Methodology section, the author has explained well but it is lengthy and some of the figures and the paragraphs are not needed. It needs to be rewritten.

The result section needs to be rewritten with more work on the discussion section.

I believe that this article has potential but there are some major changes that are needed to make this article more robust. In consequence, I recommend that the article should be considered after major changes are made. I have attached the edits on the attachment.

Reviewer 2 Report

The manuscript, A machine learning based approach for wildfire susceptibility mapping. The case study of Liguria region in Italy is a scientific and methodological work that provides interesting and innovative solutions for decision making in relation to the landscape prediction of wildfire susceptibility and ignition based in the probability for an area to burn in the future, considering the historical fire occurrence, the topographic and environmental factors influences.  The high originality of this manuscript is the machine learning tool used to predict and develop the probability ignition map.

The structure of the document is adequate and it respond to the format of a scientific article, the introduction is clear and help to understand the content of the manuscript, incorporating references and bibliographic citations of recent works in reference to the wildfires risk assessment, comparison of deterministic and stochastics approaches  and the relationships of the principal predisposing factors to fire ignition.

The writing of the different paragraphs is correct, and it is easy to read and understand the different sections, nevertheless in the section 3.2.1 “Random Forest”, would be convenient, clarify how the ML has been applicated using the variables selected in this work.

Questions and comments

1. Lines 170 to 175. The description of two models are not clear and reading the text is confused to understand each composition. I recommend insert in the table 1 two separate group of the rows, one for the “standard model” and other for the “neighboring vegetation model”

2. Line 222. Hyperparameters were set to 750 for mtree or 750 ntree?. The Random Forest bibliography recommend how a good number of n_estimators (number of trees) 100,  and indicate that higher number can be reduce the speed of process. Why the selection of n_estimators were 750?,   what was the reason for selecting this number?

3. Lines 287 to 289. A mistake in the “95thpercentile”, in the text has been wrote 47%, the correct value is 14,38% of the 5% of the area…..,

4. Line 336. Table 2. The head of the column is named “Testing BA%” but in other column is named “BurnArea (%)”. Please review this table.

5.- Line 363. The figure 5, don’t include the references (a) for map in the summer and (b) for map in the winter, and the position in the page is inverse (winter up and summer down).

6.- Lines 376 to 378. The “standard model” has been compared with the “neighboring vegetation” in the 1-fold and the 5-fold cross validation, but not with 9-fold cross validation (see table 2).

7.- Line 412. The statistical size classification denominated “large fire” is when the total burned affected is 500 ha up. Neither of the six fires mentioned in the line 408 is higher that 500 ha, because all surface of these fires, were 554 ha.  To following the standard glossary denomination of forest fires statistics, is necessary to eliminate the “large” word in this line and in the text of the figure 7.

8.- Line 414. Would be more clear,  if  the authors can insert a table showing in each one of the six fire, the pixel classification following the level susceptibility located in the surface burned, a statistic analysis comparing the fire susceptibility map with the real pixel affected can be offer the prediction capabilities.

Reviewer 3 Report

Overall I feel readers will be interested in the application of the ML method to wildfires however the writing is poor which makes it difficult to judge and evaluate the capabilities of the method.  The statistical comparisons are very difficult to follow and I am still confused about what is being represented in Table 2 and as well as Table 3?   The English writing in the paper is poor.  I've highlighted locations within the text that are written using poor grammar.  As you can see in the attached document, almost every sentence has an issue that needs to be corrected.  Note: I stopped highlighting grammatical errors halfway through the document.  I've included many questions in the attached annotated document that should be addressed by the authors.  This will require a major rewrite of the results section as well as the discussion section.  More clarity in the methods section may help readers understand the results section.  

I've provided a few suggestions in the document for improving the analysis.  This includes adding a section which documents the key factors that explained much of the spatial variability in burned area.  This would be highly useful for managers trying to understand some of the controlling factors.  I am also curious if time since the last wildfire would be an important factor if included in this analysis. 

I did not recommend rejection of the paper as I believe the content of the paper can be useful to managers and other scientists.  I feel with some major rewriting and additional analysis this could be well received.   

Round 2

Reviewer 1 Report

The authors had made significant changes in the paper. They have rewritten the abstract, Introduction, Materials and Methods along with the Result, Discussion, and Conclusion. I am pleased to see the difference in versions 1 and 2 and it is clear to read now.

The only place I felt is the Discussion section, where the authors need to bring some supporting literature. I think this paper will have a significant contribution in the wildfire community. I have attached some of my edits in the attached pdf file.

Author Response

Response to Reviewer 1 Comments

The cited report does not indicate the location of the Natura 2000 sites. The authors changed with “different Natura 2000 sites”

The authors prefers to keep the paragraph “study area” separate from “Material and method”. This structure was accepted by the two other reviewers.

The section numbering was correctly revised.

We made the revisions suggested by the anonymous reviewer on the paragraph (4.2.) “Susceptibility mapping” (see the new version of the manuscript)

In the section “Discussion”, we moved the first two lines to “Conclusions”, as suggested by the anonymous reviewer.

We added some references to support the Discussions.